# Variability of the Motor Behavior during Continued Practice of the Same Motor Game: A Preliminary Study

**Asier Gonzalez-Artetxe** [1] , **José Pino-Ortega** [2] , **Markel Rico-González** [1,2] **and Asier Los Arcos** [3,*]

[1] Department of Physical Education and Sport, University of the Basque Country UPV/EHU, 01007 Vitoria-Gasteiz, Spain; asierjfkz@gmail.com (A.G.-A.); markeluniv@gmail.com (M.R.-G.)

[2] BIOVETMED & SPORTSCI Research Group, Faculty of Sports Sciences, University of Murcia, 30720 San Javier, Spain; josepinoortega@um.es

[3] Society, Sports and Physical Exercise Research Group (GIKAFIT), Department of Physical Education and Sport, University of the Basque Country UPV/EHU, 01007 Vitoria-Gasteiz, Spain

[*] Correspondence: asier.losarcos@ehu.eus; Tel.: +34-945-013-519

**Abstract:** Motor behavior assessment during games could help physical education teachers and team coaches to design effective and efficient motor interventions. This study aimed to assess the variability of the physical and behavioral responses during continued practice of the game tail tag with a ball. Sixteen Spanish youth soccer players from an under-14 team played tail tag, with a ball, during four sessions (5 repetitions of 1 min per session). Physical (i.e., Total Distance (TD) and PlayerLoad (PL)) and behavioral (i.e., Surface Area (SA) and change in the Geometrical Centre position (cGCp)) dimensions were assessed with a local positioning system. The mean of the five series of each session was considered for further statistical analysis. The main finding was that the external load decreased ($d$ = small − large) and the use of space varied during the continued practice of tail tag. Initially, SA increased substantially ($d$ = large) and cGCp decreased slightly ($d$ = small), and then both variables tended to stabilize. This suggests that after several repetitions of the same motor game, physical education teachers and team sports coaches should use this again later, modifying this or proposing new motor games where players respond to these activities that consist in greater uncertainty than to well-known motor games.

**Keywords:** sport pedagogy; learning effects; tail tag; tactical behavior; external load

## 1. Introduction

Physical education and sport pedagogy are intervention practices that intend to modify the motor competence of students and athletes. According to the scope (e.g., school or sports academy) and the characteristics of the physical activity (e.g., soccer or basketball), the intervention can be based on different training approaches. Many of these frameworks (e.g., Teaching Games for Understanding [1,2], Play Practice [3], Game Sense [4,5], Tactical-Decision Learning Model [6], Tactical Games Approach [7], Ball School Heidelberg [8], Invasion Games Competence Model [9], Games Concept Approach [10]) use motor games as the central learning vehicle in games lessons and team sports training sessions [11] to enhance motor behavior, the whole set of observable and objectifiable motor actions performed by a student or a player [12]. Thus, the optimal design, that is, the suitable mix of structural traits [12] or constraints [13], of the games is crucial to achieving the desired effects. The structure of the games conditions the motor consequences, and consequently, the learning effects on the different dimensions of the motor behavior [12]. The value of traditional games as pedagogical tools, that foster emotional

facets or affectivity [14,15], create new relationships, develop social adaptability or boost initiative and decision, has been highlighted [16]. However, to our knowledge, the consequences of traditional motor games on other dimensions such as physical-physiological or behavioral ones have not been assessed. In order to design appropriate motor interventions based on games for performance and motor learning, sports scientists should help physical education teachers and youth sport academy team coaches assess the motor response during motor games.

Among other games, tag games are usually employed during physical education lessons or training sessions [17,18]. Compared with sports and other traditional games, these games offer different networks of motor interaction [19] in which the protagonism of each player is more than just being a member of a team [16]; for instance, one against all (e.g., classic tag game), convergent games (e.g., chain tag game), or games with permutation network (e.g., the "What's the time, Mr. Wolf?" tag game). These games stand out for enriching participants' motor experience and developing those abilities needed to later become adept team-sports players [17,20,21]. Tag games are also widely used for warm ups in physical education lessons and youth team sports training sessions [18,20]. Thus, at a practical level, it would be interesting to measure how much work was done by players during these tag games and assess the physical response between players to select optimal warm-up strategies [22].

Among different tag games, tail tag, due to its all against all structure, could be a suitable game for helping players to learn transferable tactical skills to team sports (i.e., faking or awareness of space) [17,20,21]. In addition, the original tail tag game can be modified, introducing the ball to increase the specificity for team sports players. Since planning means the periodization of the games over the school year or the sports season and the repetition of the same motor games or tasks throughout this time is common in physical education lessons and team sports academies, the assessment of the consequences due to the constant practice of the same motor games can help teachers or coaches to program motor interventions. For example, it seems that the continuous practice of a specific skill such as free-throws in basketball resulted in better technical execution [23]. But the consequences of the repetition of games consisting in more complex motor interaction networks [16,24] have not been assessed yet. Thus, the assessment of the motor behavior during the continued practice of the same tag game would help teachers and coaches to adequately use these types of motor games for a time. This means the assessment of, for example, the physical-physiological [25] and behavioral responses during the continued practice of the same tag game, as for instance tail tag. Nowadays, the external load [26] and the behavioral response can be monitored simultaneously by modern time-motion electronic performance and tracking systems (EPTS) [27].

Therefore, this study aimed to assess the variability of physical and behavioral responses during the continued practice of tail tag with a ball in youth soccer players.

## 2. Materials and Methods

### 2.1. Participants

Sixteen Spanish youth soccer players from an under-14 team (U14: $n = 16$; age: 13.0 ± 0.4 years; height: 1.56 ± 0.07 m; body mass: 46.7 ± 5.9 kg) took part in the study. The natural group was not modified for this research and all the players participated except those players (i.e., two players) that were injured a week before the assessment started. The team belongs to a soccer academy affiliated to a Spanish First Division Club (*LaLiga*). The players had about six years of playing experience and trained 32 weeks a season. The team trained twice a week during non-consecutive days on an outdoor artificial-turf field from 18:00 to 19:00. Additionally, they played an official game during the weekend at the highest competition level for their age. Their typical practice sessions consisted of a technique-based warm-up (i.e., passing drills), one or two soccer tasks (i.e., small-sided games or possession games), free play (i.e., 8-a-side game), and cooldown stretching. Parents or tutors, coaches and players, likewise the club, were fully informed of the aims and procedures of the study before giving their informed consent for the children's participation. All participants and their legal guardians

were informed about the risks and benefits and that the participants could accept and be withdrawn from the study at any time. The study protocol followed the guidelines stated in the Declaration of Helsinki (2013) and was approved beforehand by the Bioethics Commission of the University of the Basque Country (Reg. Code 132/2018).

## 2.2. Procedure

The study included four identical one-hour training sessions. The players did the same training session twice a week, with at least 48-h recovery between sessions or official matches, on the same artificial-turf field and at the same time (i.e., from 18:00 to 19:00). The training sessions started with a standardized eight-minute warm-up based on the FIFA 11+ protocol [28]. After four minutes, the players played tail tag, with a ball, five times. Each series was carried out over one minute with another minute for recovery between series. After that, they continued with their typical practice session's structure having three seven-minute soccer tasks in all training sessions in the same order, with four minutes of rest between tasks.

Tail tag is a traditional game based on capturing the tails of the other players while keeping their tail safe. Hence, the game's structure is all against all and brings about situations with ambivalent and flexible relationships [12,19]. The sixteen participants played in a 12 × 24 m rectangular field. They started the game tucking a chest guard as tail into the back of their shorts and one ball on their feet. After the playing time (i.e., a minute), the number of tails that each player had accumulated was computed. In addition, the team coach explained the game rules and ensured their accomplishment: (a) not to hold their own tail with their hands, (b) to tuck captured tails into the back of their shorts, and (c) capturing tails by only grabbing it and not the player. Each of the five series was independent, and after the recovery minute, each player started again with their tail and ball on their feet.

## 2.3. Data Collection

Data were gathered during all the series of the tail tag game during the four training sessions (i.e., five series per session × four sessions = twenty series) using a time-motion electronic performance and tracking system (EPTS) using a commercial local positioning system (LPS) (IMU; WIMU PRO^TM, RealTrack Systems, Almeria, Spain) based on ultra-wideband (UWB) technology. The UWB technology operates on a much wider frequency than other traditional radio communication technologies and a previous study did not report any problems in UWB-based tracking system accuracy in multipath conditions (i.e., 28 devices turned on) [29]. The UWB-based tracking system consists of a reference system, antennae, and tracking devices worn by all the players in a suitably fitted body vest. The antennae are transmitters and receivers of radio-frequency signals, computerizing the position of the devices that are in their coverage area, while the device receives that calculation using time difference of arrival (TDOA) [30]. This equipment and its measurements are valid, reliable for physical (e.g., distance covered) and tactical (e.g., surface area) analysis assessment [31,32] and has been awarded with the FIFA Quality Performance certificate. Moreover, data collection followed the protocol suggested by Rico-González et al. [33] on the use of technology, scoring twenty-one points out of twenty-three in their survey. Data were recorded in a training space away from metallic materials and even with low temperatures, humidity gradients and slow air circulation the conditions were maintained to allow easier positioning. The six antennae were set up around the hexagon-shaped playing field to improve signal emission and reception at a height of three meters and held by a tripod [29]. Once installed, antennae and devices were switched on one by one, afterwards waiting for five minutes to avoid technology lock [33]. In this study, LPS devices operated at a sampling frequency of 18 Hz, because low frequencies have displayed worse data quality, and 18 Hz with UWB have not shown less accuracy caused by noise problems. Data were downloaded after each session [33] and processed by S PRO ^TM software (RealTrack Systems, Almeria, Spain) [29].

## 2.4. Physical Dimension Assessment

The physical dimension was assessed with two variables: (a) Total Distance (TD), measured as meters (m) covered by the players during tail tag, and (b) PlayerLoad (PL), determined as a modified vector magnitude, expressed as the square root of the sum the squared instantaneous rate of change in acceleration in each of the three vectors—$x$, $y$ and $z$ axis—and divided by 100 [34].

## 2.5. Behavioral Dimension Assessment

Two variables were chosen to assess the behavioral dimension during tail tag: (a) Surface Area (SA), defined as total square meters ($m^2$) of a polygon described by players as its vertex point and calculated using the convex hull calculation [35], and (b) change in the Geometrical Centre position (cGCp), understood as the distance (m) between two consecutive measured points of the Geometrical Centre [36] as the mid-point of the polygon.

## 2.6. Statistical Analysis

The mean of the five series of each training session was considered for comparison between training sessions. Descriptive outcomes of TD, PL, SA and cGCp are presented as means ± standard deviations (SD). Cohen's $d$ effect size [37] was calculated to assess practical differences in motor behavior (i.e., TD, PL, SA and cGCp) between the four sessions. Effect sizes ($d$) of above 0.8, between 0.8 and 0.5, between 0.5 and 0.2, and lower than 0.2 were considered as large, moderate, small, and trivial, respectively. The coefficient of variation (CV) was used to measure the inter-player variability for the four dependent variables (i.e., TD, PL, SA and cGCp) by the formula CV = (SD · mean$^{-1}$) × 100.

# 3. Results

## 3.1. Physical Response

Except from the second to the third session ($d$ = trivial), TD and PL values decreased substantially ($d$ = small − large) over time, being the largest decrease from the first to the last session ($d$ = 1.08 − 1.22) (Figure 1).

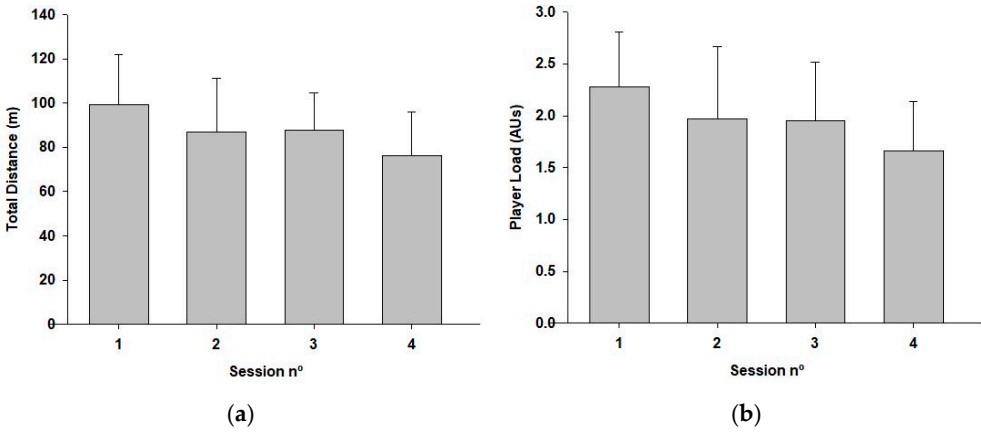

(a)                                    (b)

**Figure 1.** Physical Response: (**a**) Total Distance (m) covered by the players during the tail tag game over the four training sessions (mean ± SD per minute of play). Session 1 vs. Session 2: $d$ = 0.53, moderate; Session 1 vs. Session 3: $d$ = 0.57, moderate; Session 1 vs. Session 4: $d$ = 1.08, large; Session 2 vs. Session 3: $d$ = 0.05, trivial; Session 2 vs. Session 4: $d$ = 0.48, small; Session 3 vs. Session 4: $d$ = 0.64, moderate; (**b**) PlayerLoad (arbitrary units) accumulated during tail tag over the four training sessions (mean ± SD per minute of play). Session 1 vs. Session 2: $d$ = 0.50, moderate; Session 1 vs. Session 3: $d$ = 0.59, moderate; Session 1 vs. Session 4: $d$ = 1.22, large; Session 2 vs. Session 3: $d$ = 0.03, trivial; Session 2 vs. Session 4: $d$ = 0.53, moderate; Session 3 vs. Session 4: $d$ = 0.56, moderate.

TD inter-players variability (i.e., CV) values were 22.7%, 27.9%, 18.9% and 25.9% in the 1st, 2nd, 3rd and 4th sessions, respectively. PL CV values were 23.1%, 34.9%, 28.7% and 29.0% in the 1st, 2nd, 3rd and 4th sessions, respectively.

*3.2. Behavioral Response*

SA increased substantially from the first to the second session ($d = 1.46$, large), while the changes were trivial from the third to the fourth ($d = 0.01$). cGCp decreased slightly from the first to the second session ($d = 0.21$, small), while changes were trivial from then on (i.e., Session 2 vs. Session 3: $d = 0.01$; Session 3 vs. Session 4: $d = 0.13$) (Figure 2).

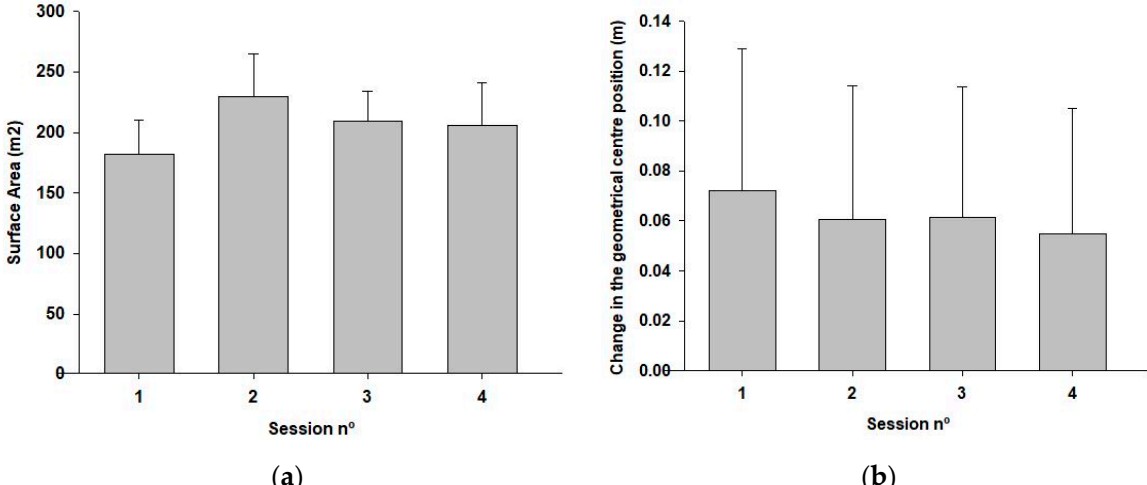

(**a**)                                                    (**b**)

**Figure 2.** Behavioral Response: (**a**) Surface Area (m$^2$) occupied by the players during tail tag over the four training sessions (mean ± SD per minute of play). Session 1 vs. Session 2: $d = 1.46$, large; Session 1 vs. Session 3: $d = 1.00$, large; Session 1 vs. Session 4: $d = 0.75$, moderate; Session 2 vs. Session 3: $d = 0.65$, moderate; Session 2 vs. Session 4: $d = 0.66$, moderate; Session 3 vs. Session 4: $d = 0.01$, trivial; (**b**) change in the Geometrical Centre position (m) during tail tag over the four training sessions (mean ± SD per minute of play). Session 1 vs. Session 2: $d = 0.21$, small; Session 1 vs. Session 3: $d = 0.20$, small; Session 1 vs. Session 4: $d = 0.33$, small; Session 2 vs. Session 3: $d = 0.01$, trivial; Session 2 vs. Session 4: $d = 0.12$, trivial; Session 3 vs. Session 4: $d = 0.13$, trivial.

SA variability (i.e., CV) values were 15.5%, 15.6%, 12.1% and 16.9% in the 1st, 2nd, 3rd and 4th sessions, respectively. cGCp CV values were 78.7%, 87.8%, 85.9% and 92.0% in the 1st, 2nd, 3rd and 4th sessions, respectively.

## 4. Discussion

The aim of the study was to assess the variability of physical and behavioral responses during the continued practice of tail tag with a ball in youth soccer players. The main finding was that the continued practice of the same motor game (i.e., tail tag) meant a decrement of the external load (i.e., TD and PL) and different uses of space over time. Specifically, the polygon described by the players (i.e., SA) was greater, while the change of the mid-point of this polygon (i.e., cGCp) decreased.

As we have mentioned, tag games are usually used to warm up during physical education lessons or team sports training sessions [18,20]. Thus, these games would prepare the student or the player for the session respect to the dimensions of motor competence (e.g., physical-physiological, behavioral and emotional-affective dimensions) [22,38,39]. Regarding the physical-physiological dimension, it was found that the physical response varied considerably between sessions (Figure 1), that is, the same tag game can induce very different physical efforts during time. Specifically, the constant repetition of the motor game meant a constant substantial decrement of the external load (Figure 1). Since tail

tag did not ensure a similar physical effort during the sessions, the use of these motor games is not suggested to warrant optimal physical stimulus for attaining optimal performance [22] in the next drills/tasks of the session. Additionally, the CV (i.e., inter-player variability) ranged from the 18.9 to 27.9% for the TD and from 23.1 to 34.9% for the PL, indicating that the same tag game means very different physical response between players. Similarly, the CV ranged from 11.9 to 40.5% for the heart rate-based training load, overall perceived and muscular perceived efforts during warm-up composed of a technical drill (i.e., collaboration drill, 8 vs. 0 players) in youth soccer players [25]. It seems that these types of tasks did not ensure a similar physical effort in the players, suggesting that the condition of each player for the next drill/task differs considerably at the physical-physiological level. Thus, a controlled and individualized physical exercise would be more adequate than tag games and technical drills to ensure a specific physical-physiological level, but other dimensions as, for example, decision making, behavioral and technical dimensions, would not be required.

Regarding the behavioral response during tail tag, a similar trend was observed for both variables (i.e., SA and cGCp) over the four training sessions. Initially, SA increased substantially ($d = 1.46$, large) and cGCp decreased slightly ($d = 0.21$, small) from the first to the second session (Figure 2). Then, both variables (i.e., SA and cGCp) tended to stabilize (i.e., trivial changes between the last sessions) (Figure 2). The greater space occupied by the participants suggested that they had adopted a defensive behavior, making it more difficult to capture the tails of the other players, but easier in keeping their tail safe. As well, the primary decrease and the posterior stabilization of cGCp indicated that the changes of the mid-position of the group of players are more stable from session to session. Another reason for the stabilization of the use of space can be non-aggression pacts between players. The game's structure (i.e., all against all) bring about situations with ambivalent and flexible relationships [12,19]. These results suggested that the game itself, without any external instructions, lead players to a more stable use of space after the continued practice of the same motor game (i.e., tail tag). Thus, if the purpose of physical education teachers or team coaches is to demand students' or players' continuous adaptability to face up to diverse and changing motor situations [40–42], the constant practice of the same motor game might not be an appropriate learning strategy. Nevertheless, it would be interesting to develop similar interventions based on tag games over longer periods between sessions and in different educational or sports contexts in the future, so more could be learnt about the possible retention and transfer effects [8,43,44] of these motor games.

## 5. Conclusions

Tail tag does not result in a similar external load between players from session to session. The continued practice of the same tag game (i.e., tail tag) has, consequently, the stabilization of the motor behavior of the group of players and the reduction of physical effort. This suggests that after several repetitions of the same motor game, physical education teachers and team sports coaches should use this again later, modifying it or proposing new motor games where players respond to these activities that consist in greater uncertainty than to well-known motor games.

**Author Contributions:** Conceptualization, A.G.-A., A.L.A., and M.R.-G.; methodology, A.G.-A., A.L.A., and M.R.-G.; formal analysis, A.G.-A., A.L.A., and J.P.-O.; investigation, A.G.-A., A.L.A.; resources, A.G.-A., A.L.A., and J.P.-O.; data curation, A.L.A., J.P.-O., and M.R.-G.; writing—original draft preparation, A.G.-A. and A.L.A.; writing—review and editing, A.G.-A. and A.L.A.; supervision, A.L.A., A.G.-A., J.P.-O. and M.R.-G.; project administration, A.L.A., A.G.-A.; funding acquisition, A.L.A. All authors have read and agreed to the published version of the manuscript.

**Funding:** The authors gratefully acknowledge the support of a Spanish government subproject Integration ways between qualitative and quantitative data, multiple case development, and synthesis review as main axis for an innovative future in physical activity and sports research [PGC2018-098742-B-C31] (Ministerio de Ciencia, Innovación y Universidades, Programa Estatal de Generación de Conocimiento y Fortalecimiento Científico y Tecnológico del Sistema I + D + i), that is part of the coordinated project New approach of research in physical activity and sport from mixed methods perspective (NARPAS_MM) [SPGC201800X098742CV0].



**Acknowledgments:** The authors thank the players and staff of the C.D. Pamplona soccer academy for participating and cooperating in this project.

**Conflicts of Interest:** The authors declare no conflict of interest. The funders had no role in the design of the study; in the collection, analyses, or interpretation of data; in the writing of the manuscript, or in the decision to publish the results.

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
