# Peer review of "Variability of the Motor Behavior during Continued Practice of the Same Motor Game: A Preliminary Study"

_sustainability, doi:10.3390/su12229731_

Round 1

Reviewer 1 Report

This manuscript aimed to assess the variability of the physical and behavioural responses during continued practice of the game tail tag with a ball in youth soccer players. The manuscript is very well written, the rationale is clear, and the discussion & conclusion sections are mostly appropriate given the results. However, I have some issues that I believe need to be addressed largely related to the statistical analysis.

  1. Line 121-123: Has the equipment been validated to assess all of the measures used in the current study?
  2. Line 147: How was normality of the data assessed?
  3. Line 149-154: Why wasn’t a repeated measures ANOVA performed? Why not use null hypothesis significance tests?
  4. Line 149-150: I recommend including 95% CI for effect sizes. The CI around the small effects are presumably very large?
  5. Cohen’s d is not appropriate for small sample sizes, Hedges’ g should be used instead.
  6. Figures: Resolution needs to be improved.
  7. Line 215: How confident are you that the small effect was real? Given the relatively small sample size your estimates will not be very precise.
  8. Line 233: Consider rephrasing as “Tail tag does not result in a similar…””

Reviewer 2 Report

Congratulations on the work done

Keywords: Authors should decrease the number of keywords and be more specific

Introduction: In the introduction, it is important to address the main results of studies that work with the variability of physical and behavioral responses, as well as factors that affect this type of variables in the studied population.

There must be evidence of the effects of the type of games applied in this study.

Methods: the inclusion and exclusion criteria used in the sample selection must be clarified.

Round 2

Reviewer 2 Report

Congratulations on the work done